# Flood loss modelling with FLF-IT: a new Flood Loss Function for Italian residential structures

Roozbeh Hasanzadeh Nafari[1,2], Mattia Amadio[3,4], Tuan Ngo[5] and Jaroslav Mysiak[3,4]

[1]Centre for Disaster Management and Public Safety (CDMPS), Department of Infrastructure Engineering, University of Melbourne, Melbourne, 3010, Australia
[2]Bushfire & Natural Hazards Cooperative Research Centre, Melbourne, 3002, Australia
[3]Fondazione Eni Enrico Mattei (FEEM) and Euro-Mediterranean Centre on Climate Change (CMCC), San Giorgio Maggiore, Venice, Italy
[4]Department of Economics, Ca' Foscari University, Venice, Italy
[5]Director of the Advanced Protective Technologies for Engineering Structures (APTES) Group, Department of Infrastructure Engineering, University of Melbourne, Melbourne, 3010, Australia

*Correspondence to*: R. Hasanzadeh Nafari (rhasanzadeh@student.unimelb.edu.au)

**Abstract.** The damage triggered by different flood events costs the Italian economy millions of Euros each year. This cost is likely to increase in the future due to climate variability and economic development. In order to avoid or reduce such significant financial losses, risk management requires tools which can provide a reliable estimate of potential flood impacts across the country. Flood loss functions are an internationally accepted method for estimating physical flood damage in urban areas. In this study, we derived a new Flood Loss Function for Italian residential structures (FLF-IT), on the basis of empirical damage data collected from a recent flood event in the region of Emilia-Romagna. The function was developed based on a new Australian approach (FLFA), which represents the confidence limits that exist around the parameterized functional depth-damage relationship. After model calibration, the performance of the model was validated for the prediction of loss ratios and absolute damage values. It was also contrasted with an uncalibrated relative model with frequent usage in Europe. In this regard, a three-fold cross-validation procedure was carried out over the empirical sample to measure the range of uncertainty from the actual damage data. The predictive capability has also been studied for some sub-classes of water depth. The validation procedure shows that the newly derived function performs well (no bias and only 10% mean absolute error), especially when the water depth is high. Results of these validation tests illustrate the importance of model calibration. The advantages of the FLF-IT model over other Italian models include calibration with empirical data; consideration of the epistemic uncertainty of data; and the ability to change parameters based on building practices across Italy.

## 1 Introduction

Floods are the natural hazards that cause the largest economic impact in Europe today (European Environment Agency, 2010). Italy is no exception, with about 80% of its municipalities being exposed to some degree of hydrogeological hazards (Zampetti et al., 2012). Regarding flood hazard frequency, 8% of Italy's territory and 10% of its population are exposed to a flood probability of once every 100 to 200 years (ANCE/CRESME, 2012; Trigila et al., 2015). This issue is reflected in over a billion Euros spent from 2009 to 2012 on recovery from extreme hydrological events (Zampetti et al., 2012). Italy is, in fact, the European country where floods generate the largest economic damage per annum (Alfieri et al., 2016). This is especially worrisome considering that the frequency of extreme flood losses may be doubled at least by 2050 in Europe due to climatic change factors and urban expansion (Jongman et al., 2014). Climate variability already affects rainfall extremes and the peak volumes of discharge in rivers (Alfieri et al., 2015; Karagiorgos et al., 2016). Relentless urban sprawl within catchments alters the water run-off speed and propagation while increasing the value of exposed land use (Barredo, 2009). In order to effectively prevent massive losses, disaster risk management requires estimation well in advance of the frequency and magnitude of potential flood events, and their consequences in terms of economic damages (Elmer et al., 2010; Hammond et al., 2015; Kaplan and Garrick, 1981; Neale and Weir, 2015; Thieken et al., 2008; UNISDR, 2004). Therefore, it is indispensable to provide decision-makers with reliable assessment tools that are able to produce such knowledge, after which an efficient risk reduction strategy can be adequately planned (Emanuelsson et al., 2014; McGrath et al., 2015; Merz et al., 2010; Penning-Rowsell et al., 2005).

In general, flood losses are classified as marketable (tangible) or non-marketable (intangible) values, and as direct or indirect (Jonkman, 2007; Kreibich et al., 2010; Meyer et al., 2013; Molinari et al., 2014a; Thieken et al., 2005). Direct damage takes place when the floodwater physically inundates buildings and structures, whereas indirect damage accounts for the consequences of direct damage on a wider scale of space and time (Hasanzadeh Nafari et al., 2016c). The tools employed to assess flood risk consist of a variety of damage models, with differing methods depending on the type of accounted losses. While Input-Output models, Computable General Equilibrium models and other econometric tools are often used to estimate indirect economic losses (Carrera et al., 2015; Hallegatte, 2008; Koks et al., 2015), the focus of most flood damage models is still on the estimation of direct, tangible losses using stage-damage curves. Stage-damage curves or flood loss functions are used to depict a relationship between water depth and economic damage for a specific kind of structure or land use (Jongman et al., 2012; Kreibich and Thieken, 2008; Merz et al., 2010; Messner et al., 2007; Thieken et al., 2009). Damage curves can be empirical or synthetic. Empirical curves are drawn based on actual data collected from one specific event. Due to the differences in flood and building characteristics, they cannot be directly employed in different times and places (Gissing and Blong, 2004; McBean et al., 1986). To resolve this issue, general synthetic curves based on a valuation survey have been created for different types of buildings. Valuation surveys assess how the structural components are distributed in the height of a building (Barton et al., 2003; Smith, 1994). Afterwards, the magnitude of potential flood losses is estimated based on the vulnerability of structural components and via "what-if" questions (Gissing and Blong, 2004; Merz et al., 2010). Damage

functions can also be distinguished as absolute or relative. The first type states the damage directly in monetary terms, while the relative type states the damage as a percentage of the total exposed value, which can refer to the total replacement value or the total depreciated value (Kreibich et al., 2010). Relative functions have an advantage over absolute functions, namely that they are more flexible for transfer to different regions or years since the damage ratio is independent of the changes in

market values (Merz et al., 2010). Still, both types are developed on sample areas which have particular geographical characteristics that affect both the quality of the exposed value and the flood phenomena (McGrath et al., 2015; Proverbs and Soetanto, 2004). Therefore, transferred models may carry a high level of uncertainty, unless they are calibrated with an empirical dataset collected from the new study area (Cammerer et al., 2013; Hasanzadeh Nafari et al., 2015; Molinari et al., 2014b).

Although Italy has seen several flood disasters in recent years, flood records do not enable development or validation of a national loss flood function because the information is still poor, fragmented and inconsistent. This issue largely depends on the lack of an established official procedure for the collection and the storage of damage data (Molinari et al., 2014b). Another obstacle is the heterogeneity across different regions of digital geographic information, which is the key to correctly represent the driving factors of exposure and vulnerability influencing the sustained damage. Few attempts at drawing a

depth-damage relation from post-disaster reports have been made (Amadio et al., 2016; Luino et al., 2009; Molinari et al., 2014b, 2012; Papathoma-Köhle et al., 2012; Scorzini and Frank, 2015), while other uncalibrated synthetic functions have been derived from pan-European studies (Huizinga, 2007). The use of such uncalibrated functions on the Italian territory has proven troublesome (Amadio et al., 2016), showing a large degree of uncertainty.

Our research aims to calibrate and validate a new relative flood loss function for Italian residential structures (FLF-IT) based

on real damage data collected from one large river flood event in the region of Emilia-Romagna at the beginning of 2014. The focus of this study is on direct tangible damage, and the spatial scale is on the order of individual buildings. This research builds on a newly derived Australian approach called FLFA (Hasanzadeh Nafari et al. 2016a, 2016b).

## 2 Case study

The region of Emilia-Romagna is located in Northern Italy, on the southern side of the Po River, the longest of all Italian

rivers. This region has the greatest flood prone area both in relative and absolute terms: about 10,000 km$^2$, including 64% of the population are exposed to a medium flood probability (return period between 100 and 200 years), while 2,500 km$^2$ and 10% of the population are exposed to a high probability (return period between 20 and 50 years) (Trigila et al., 2015). This includes more than half of the region's territory. Our empirical data comes from a flood generated by the Secchia river in 2014 near the town of Modena, in the central part of Emilia-Romagna.

## 2.1 Event description

January 2014 was a dramatic month for floods in Italy, with 110 flood events recorded over a span of 23 days due to extreme meteorological conditions. Severe precipitations hit central Emilia-Romagna between the 17[th] and the 19[th] of January, with an areal mean of 125 mm of cumulative rain over 72 hours flowing in the Secchia catchment. The increase in the river flow volumes caused heavy stress on the levees, which stand 7-8 meters over the flood plain. At around 6 am, approximately 10 meters of the eastern Secchia levee were overwashed and breached at the top by one meter, thereby starting to flood the countryside. In 9 hours, the levee section was completely destroyed for a length of 80 meters, spilling 200 $m^3$ per second in the surrounding plain and flooding nearly 65 $km^2$ of rural land (Figure 1) (D'Alpaos et al., 2014). Seven municipalities have been affected, with the small towns of Bastiglia and Bomporto suffering the largest share of losses. Both towns, including their industrial districts, remained flooded for more than 48 hours. The total volume of water inundating the area was estimated to be around 36 million $m^3$ (D'Alpaos et al., 2014).

## 2.2 Data description

The information about cumulative water depths comes from the hydraulic simulation of the event produced by the technical-scientific committee in the official report (D'Alpaos et al., 2014; Vacondio et al., 2014). The extent of the simulated flood is nearly 5 $km^2$, with an average depth of one meter. The flow volume at the breach is calculated using the 1-D model HEC-RAS calibrated on recorded observations from the event. The evolution of the flooding is simulated by a 2-D hydraulic model using the finite-volume method over a Digital Terrain Model (DTM) obtained by LiDAR scans at a one-meter resolution. The simulation also accounts for the gradual change in the size of the breach from 10 to 80 meters (Vacondio et al., 2014).

A database of damage declared by residential properties has been made available for this research by the local authorities. Damage records are listed by address for the three municipalities of Bastiglia (70% of the total damage), Bomporto (24%) and Modena (6%). The total damage sums up to EUR 41.5 million, of which: 54% is damage to structural parts, including installations; 33% is damage to movable contents, meaning furniture and common domestic appliances; and 13% is represented by registered vehicles, such as cars and motorcycles. For the purpose of our study, only the structural damage is considered. The recorded damage is compared to the average market values of the residential properties, as reported by the cadastral map for the semester preceding the flood event (Agenzia delle Entrate, 2014). The majority of residential structures in the area share the same general characteristics: they are brick or concrete buildings built in the last 30 years, with no underground basement or parking (slab-on-ground). Houses have at least two or three floors. However, only the ground floors have been affected in this particular event.

The information related to water depth, total structural damage and average market value is linked together at the building scale (Fig. 2) by combining the street numbers points and residential buildings perimeters from the official regional geodatabase (Regione Emilia Romagna, 2011). The mean of cumulative water depths simulated by the hydraulic model is

calculated within the area of each building unit. Accordingly, each address linked to a damage record is first georeferenced as a street number point; then the points falling within the same building unit are summed into an aggregated value representing the total structural damage occurred in that building, including private dwellings and common parts. This spatial join is necessary since building perimeters do not include any information about addresses. The procedure is performed

successfully for EUR 21.7 million, corresponding to 97% of the total residential damage. The remaining 3% of records are excluded due to incomplete addresses or inconsistency with the spatial data. Percentages of damage vs. depths of water for all 613 final samples have been depicted in Fig. 3.

## 3 The FLFA method

The FLFA method is based on a simplified synthetic approach called the sub-assembly method, proposed by the HAZUS

technical manual (FEMA, 2012). This method measures the extent of losses for each stage of floodwater and suggests a flexible curve that accounts for the variability in the characteristics of structures. In the first step, one or more representative building categories are selected from the study area. The ratio of damage for every stage of water and within each category of the building is a function of the vertical distribution of structural components (i.e., vulnerability and the total value exposed to flood) (Lehman and Hasanzadeh Nafari, 2016). More specifically, each structural component starts suffering

damage after a specific stage is reached. Commonly the first decimetres of water cause damage to some of the most valuable items such walls, floors, insulation and electrical wiring (FEMA, 2012). Accordingly, the relationship between the damage percentage ($d_h$) and water depth can be described by a root function (Cammerer et al., 2013; Elmer et al., 2010; Kreibich and Thieken, 2008). The following function (1) is developed by Hasanzadeh Nafari et al. (2016a) for the Australian case study:

$$d_h = \left(\frac{h}{H}\right)^{\frac{1}{r}} \times D_{max} \qquad\qquad (1)$$

The root ($r$) controls the rate of alteration in the percentage of damage relative to the growth of the water depth ($h$) over a total height ($H$) of the floor. The $D_{max}$ is the total percentage of damage corresponding to the total height of the floor. A higher value of $r$ means a slower increase in the rate of damage. The obtained curve is then adjusted and calibrated using the empirical data collected from the selected study area. Hence, this approach is defined as an empirical-synthetic method. Due to the inherent uncertainty in the data sample, the study has employed a bootstrapping approach, which produces three stage-

damage functions (i.e. most likely, maximum and minimum damage functions) for each type of building. This range of estimate describes confidence limits around the functional parameters and represents the uncertainty that exists in the data sample. The advantages of this simplified synthetic approach include calibration with empirical data, a better level of transferability in time and space, consideration of the epistemic uncertainty of data, and the ability to change parameters based on building practices across the world.

## 4 Calibration of FLF-IT

Based on the formula represented previously, the model calibration process includes choosing the most appropriate values for the root of function and the maximum percentage of damage (i.e., $r$ and $D_{max}$), with reference to the empirical dataset (Hasanzadeh Nafari et al., 2016a). The selection will be made by the chi-square test of goodness of fit, to minimise predictive errors. Also, instead of a deterministic regression analysis, this study has relied on the probabilistic relationship among the percentage of damage and other damage-related parameters (i.e. building and flood characteristics) (Hasanzadeh Nafari et al., 2016b). In this regard, a bootstrapping approach has been employed to resample the damage data 1,000 times. This method assists in exploring the confidence limits around the parameters and illustrates the epistemic uncertainty of the empirical damage data (Lehman and Hasanzadeh Nafari, 2016). To be more specific:

- First, the original dataset including 613 data points was resampled using a bootstrapping approach;

- For the new resample, the most appropriate value of the root function and the maximum percentage of damage were selected by the chi-square test of goodness of fit;

- The two previous steps were repeated 1,000 times, and 1,000 sets of parameters (i.e., $r$ and $D_{max}$) were generated as the result;

- Finally, by the above iteration, the averages of the 1,000 calibrated parameters converged to a fixed value considered as the most likely scenario. The most likely parameters produce the smallest cumulative error compared to the actual damage data.

- Also, from the 1,000 sets of parameters generated above, the function that maximises the depth-damage relationship was taken as a maximum damage curve, and the observation that created the minimum depth-damage relationship was considered for the minimum depth-damage function.

Results of the model calibration are presented in Table 1 and Fig. 4.

**Table 1. Number of samples and range of r and D$_{max}$ values, calculated by the bootstrap and chi-square test goodness of fit.**

| Number of Samples | Parameters | Range of parameters | | |
|---|---|---|---|---|
| | | Minimum | Most likely | Maximum |
| 613 | r | 2.7 | 2 | 1.7 |
| | $D_{max}$ | 10% | 20% | 40% |

## 5 Model validation

### 5.1 Applied damage models

Besides FLF-IT, Damage Scanner as an uncalibrated relative model with frequent usage in Europe has been selected for comparison in this study. The Damage Scanner model (de Bruijn, 2006; Klijn et al., 2007) is based on depth-damage curves

previously developed by the synthetic approach in the Netherlands using data from "what-if" analyses at the building scale (Kok et al., 2004). These curves estimate the magnitude of damage separately for building structure and movable content. The damage is expressed in relation to an average maximum damage value per square meter, which varies according to land use classes (e.g. residential, industrial, agriculture, and infrastructure). The Damage Scanner model have been employed for predictive purpose in various studies (Aerts and Botzen, 2011; Bouwer et al., 2010; de Moel et al., 2011; Koks et al., 2012;

Poussin et al., 2012; Ward et al., 2011), and it has been more recently updated including additional land use subclasses (de Moel et al., 2013; Koks et al., 2014). The uncertainty of Damage Scanner has been investigated in comparison to other damage models (Bubeck et al., 2011; Jongman et al., 2012), and its transferability has been evaluated for use in different areas of study such as Northern Italy (Amadio et al., 2016). Damage Scanner is, in fact, easy to tailor to land use description available for Italy, and because it expresses damage in relative terms, it can be adapted to work on region-specific maximum

values. For the purpose of comparison with FLF-IT, the curve related to residential structure damage has been selected from the Damage Scanner set and applied at building scale on the residential units using the same average market values and simulated water stages employed to produce the FLF-IT. It is worth noting that the predicted absolute damage values are calculated by multiplying the estimated loss ratio by the average market value and the area of each property.

### 5.2 Result comparison and model validation

Results of the applied damage models have been compared with the observed loss data, and their performances have been validated in contrast to real damage data. Due to the lack of an independent dataset, a three-fold cross-validation technique was employed for this purpose (Seifert et al., 2010). Accordingly, the original damage records including 613 data points were first shuffled and partitioned into three equally sized subsets. Then, three iterations of model calibration and model testing were performed. In each iteration, one subset including 204 samples was singled out for model testing, while the

remaining two parts including 409 data points were used for model calibration (Refaeilzadeh et al., 2009). Model calibration in each iteration was performed based on the approach explained earlier. Eventually, the loss ratio of the held-out subset was estimated by the FLF-IT model calibrated without it, and the results were compared with the actual records. Errors including the mean bias error (MBE), the mean absolute error (MAE) and the root mean square error (RMSE) were calculated and averaged over all three iterations. The MBE illustrates the direction of the error bias (i.e. a positive MBE shows an

overestimation in the predicted values, while a negative MBE depicts an underestimation); the MAE shows how close the estimates are to the actual damage ratios; and the RMSE signifies the variation of the predicted ratios from the actual records

(Chai and Draxler, 2014; Seifert et al., 2010). In addition to FLF-IT and for each iteration, errors of the Damage Scanner model's estimates were calculated. The results are presented in Table 2.

This table clearly shows that FLF-IT has a better performance compared to the Damage Scanner model which is not calibrated with the local damage data. The average of the MBE over all iterations shows no bias and represents only around 1% bias in each iteration. The MAE is 10% on average, and RMSE alters between 12 and 16% (14% on average). The results of the Damage Scanner model show 13% average deviation from the validation subsets ratios; larger average values of absolute error; and higher variation of the predicted ratios from the actual records. Overall, the small value of the deviations and the low variation of the errors signify that the new model performance is accurate.

**Table 2. Error estimation for the performance of the FLF-IT model (MBE: Mean Bias Error; MAE: Mean Absolute Error; RMSE: Root Mean Squared Error).**

| | MBE | | MAE | | RMSE | |
|---|---|---|---|---|---|---|
| | **FLF-IT** | **Damage Scanner** | **FLF-IT** | **Damage Scanner** | **FLF-IT** | **Damage Scanner** |
| **Iteration 1** | 0.015 | 0.152 | 0.092 | 0.188 | 0.119 | 0.212 |
| **Iteration 2** | -0.010 | 0.125 | 0.104 | 0.177 | 0.157 | 0.204 |
| **Iteration 3** | -0.009 | 0.125 | 0.091 | 0.164 | 0.133 | 0.188 |
| **Average** | 0.00 | 0.13 | 0.10 | 0.18 | 0.14 | 0.20 |

The predictive capability has also been studied for some sub-classes of water depth. By this test, the performance of the applied damage models will be evaluated for different stages of the flood. Figs.5 and 6 show the precision of the results and the number of relative damage records for seven different sub-classes of water depth. These Figures clearly show that the uncertainty of FLF-IT is less than the Damage Scanner model and the results justify the overall better performance of the FLF-IT model. This test shows that the application of the Damage Scanner model using the original uncalibrated maximum damage values leads to overestimating the actual damage occurred during this flood event especially when the water depth is high. In contrast to Damage Scanner, FLF-IT performs well specifically when the flood is deep, the extent of damage is more considerable, and the prediction performance of the model is more important. The high number of samples with a depth more than 60 centimetres supports the reliability of this outcome.

In addition to the above comparison on the loss ratios, the performance of the model is also validated for predicting the absolute damage values. As stated before, the overall reported loss for the 613 cases (building fabric) amounted to EUR 21.7 million. In this regard and for each iteration, the absolute damage records are resampled using the bootstrapping approach 10,000 times, and the 95% confidence interval of the total losses was calculated. If the total damage value estimated by the models falls within the 95% confidence interval, their performance is accepted. Otherwise, it is rejected (Cammerer et al.,

2013; Seifert et al., 2010; Thieken et al., 2008). By this approach, the performance of the applied damage models in terms of structural damage estimation in the area of study will be evaluated. The results are presented in Table 3, which shows that the results of all iterations of the FLF-IT model with the most likely functional parameters $r$ and $D_{max}$ lie within the 95% confidence intervals and the FLF-IT model has an acceptable performance. However, results of Damage Scanner do not lie

5   within the confidence intervals of the mean loss ratios, and its performance is rejected in this area of study. Fig. 7 represents the workflow and the methodological steps of this study.

Results of these validation tests illustrate the importance of model calibration, especially when the water depth is the only hydraulic parameter taken into account (Cammerer et al., 2013; Chang et al., 2008; McBean et al., 1986). In other words, flood damage, being a complicated process, could be dependent on more damage influencing parameters than those

10   considered here (Fuchs et al., 2011; Grahn and Nyberg, 2014; Hasanzadeh Nafari et al., 2016c; Merz et al., 2013; Schröter et al., 2014). However, by calibrating the loss function with an actual damage dataset and providing an empirically-based model, the function estimations are good (i.e. low predictive error, low variation and acceptable reliability in results) and its performance is validated for use in flood events with the same geographical conditions (i.e. flood characteristics and building specifications) as the area of study (Hasanzadeh Nafari et al., 2016b; McBean et al., 1986).

**Table 3. Comparison of total absolute losses estimated by FLF-IT with the 95% confidence interval of the resampled damage records.**

| | 95% confidence interval | Estimated damage values (in 10^6 EUR) | | | |
| --- | --- | --- | --- | --- | --- |
| | | **FLF-IT** | **Within 95% interval** | **Damage Scanner** | **Within 95% interval** |
| **Iteration 1** | 4.88-6.8 (2.5th-97.5th percentile) | 6.5 | Yes | 16.2 | No |
| **Iteration 2** | 5.81-7.8 (2.5th-97.5th percentile) | 7.7 | Yes | 15.6 | No |
| **Iteration 3** | 8.07-10.4 (2.5th-97.5th percentile) | 10.1 | Yes | 21.8 | No |
| **All records** | 19.94-24.5 (2.5th-97.5th percentile) | 24.3 | Yes | 53.7 | No |

While the FLF-IT model is shown to be more accurate, there are still some limitations that can be the subject of new

20   research. Model validation in this study was based on random samples which were not independent of the data used for model calibration, and this test does not give information about the transferability of the FLF-IT model. Hence, improvements can be made by considering more influencing factors of hazard, exposure and vulnerability; validation with more actual damage records from other study areas in Italy; and considering other types of structure.

# 6 Conclusion

Floods are frequent natural hazards in Italy, triggering significant negative consequences on the economy every year. Their impact is expected to worsen in the near future due to socio-economic development and climate variability. To be able to reduce the probability and magnitude of expected economic losses and to lessen the cost of compensation and restoration,
flood risk managers need to be correctly informed about the potential damage from flood hazards on the territory. A loss function that can reliably estimate the economic costs based on available data is the key to achieving this objective. However, despite a significant number of flood disasters hitting Italy every year, few attempts at developing a flood damage model from post-disaster reports have been made.

Flood loss functions are an internationally accepted method for estimating direct flood damage in urban areas. Flood losses
can be classified as marketable or non-marketable values, and as direct or indirect damages. This study focused on direct, marketable damage due to riverine floodwater inundation. We employed a newly derived Australian approach (FLFA) with empirical damage data from Italy to develop a synthetic, relative flood loss function for Italian residential structures (FLF-IT). The FLFA approach takes data of damage and depth, stratified by building classifications, and uses the chi-square test of goodness of fit to fix a parameterized function to compute depth-damage estimates. Parameters include the height of the
stories, maximum damage as a percentage of the total building value, and the elevation of water which building start damaging. Additionally, FLFA illustrates a bootstrapping approach to the empirical data to assist in describing confidence limits around the parameterized functional depth damage relationship. Accordingly, the advantages of the new model (FLF-IT) include calibration with empirical data, consideration of the epistemic uncertainty of data and the ability to change parameters based on building practices across Italy. After model calibration, its performance was also validated for
predicting the loss ratios and absolute damage values. Also, the performance of the new model in comparison to the empirical data has been contrasted with an uncalibrated relative model with frequent usage in Europe. In this regard, a three-fold cross-validation procedure and the usual bootstrap approach were applied to the empirical sample to measure the range of uncertainty from the actual damage data. This validation test was selected to compensate for the lack of comparable data from an independent flood event. Finally, the predictive capability has also been studied for some sub-classes of water depth.
The validation procedure shows that estimates of FLF-IT are good (no bias, 10% mean absolute error and 14% root mean square error) especially when the flood is deep, and its performance is acceptable. However, the application of the Damage Scanner model using the original uncalibrated maximum damage values leads to overestimating the actual damage occurred during this flood event.

Results of these validation tests depict the importance of model calibration, especially when the water depth is the only
hydraulic parameter considered. In other words, by calibrating the loss function and providing an empirically-based model, the function performs well (i.e. low predictive error, low variation and acceptable reliability) and its performance is validated for use in events with the same geographical conditions as the area of study. Awareness of these issues is necessary for decision-making in flood risk management. Further research will be aimed at considering some additional parameters that

may govern the significance of the damages for a given depth. An independent dataset is required to evaluate the predictive capacity and transferability of the model.

**Acknowledgment**

The authors would like to acknowledge the ongoing financial contribution and support from the Bushfire & Natural Hazards Cooperative Research Centre. The research leading to this paper has also received funding from project ENHANCE (Grant Agreement N° 308438) and the People Programme (Marie Curie Actions) of the European Union's Seventh Framework Programme FP7 (Grant Agreement number 609642). The authors want to thank the local administrations of Emilia-Romagna for providing the official damage records of the 2014 flood event.

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

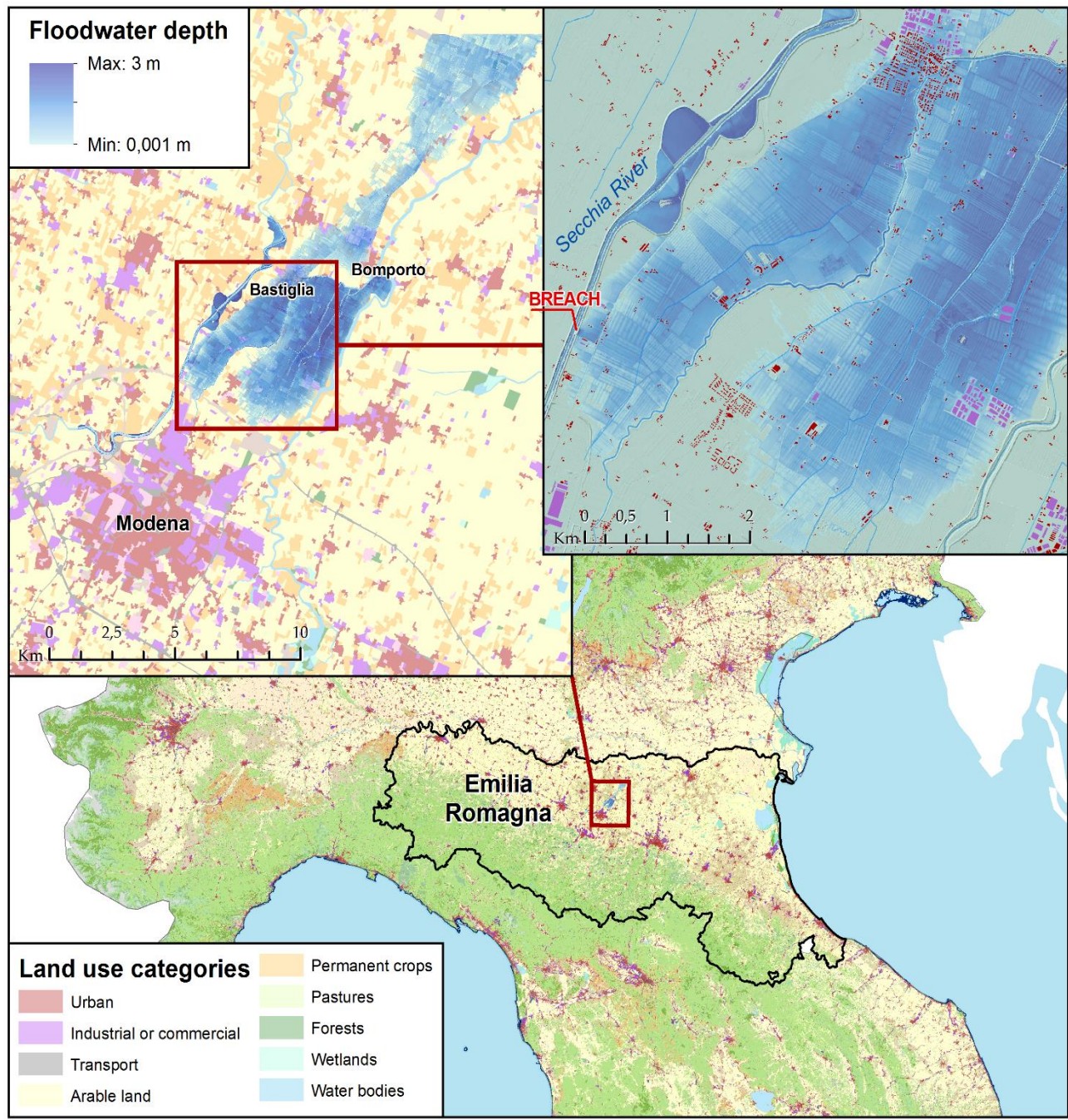

**Figure 1. Identification of case study, flooding from the river Secchia during January 2014 in central Emilia-Romagna, Northern Italy.**

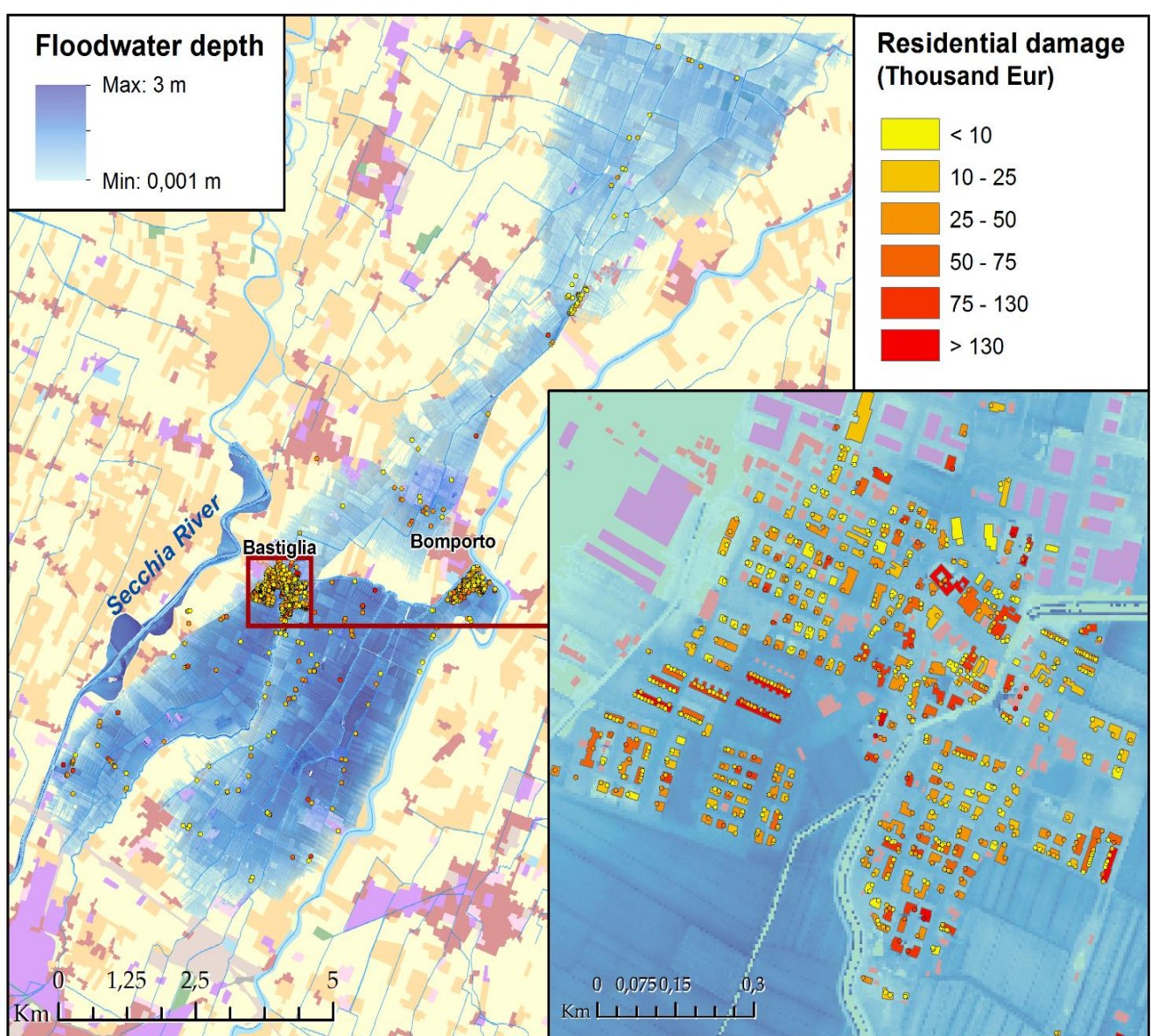

**Figure 2. Visualisation of the empirical damage records suffered by the individual dwellings during the flood event of 2014. Records are projected to official street number points by using their "address" field. The information is then transferred from the points to the building features that contains them. The point records that fall within the same building perimeter are summed up into one aggregated damage value for each residential building. About 97% of damage records are correctly projected. The remaining 3% of damage records is discarded due to inconsistent projection, incomplete address or gaps in the record data. The colour gradient (yellow to red) indicates the magnitude of the damage for both individual points and building units.**

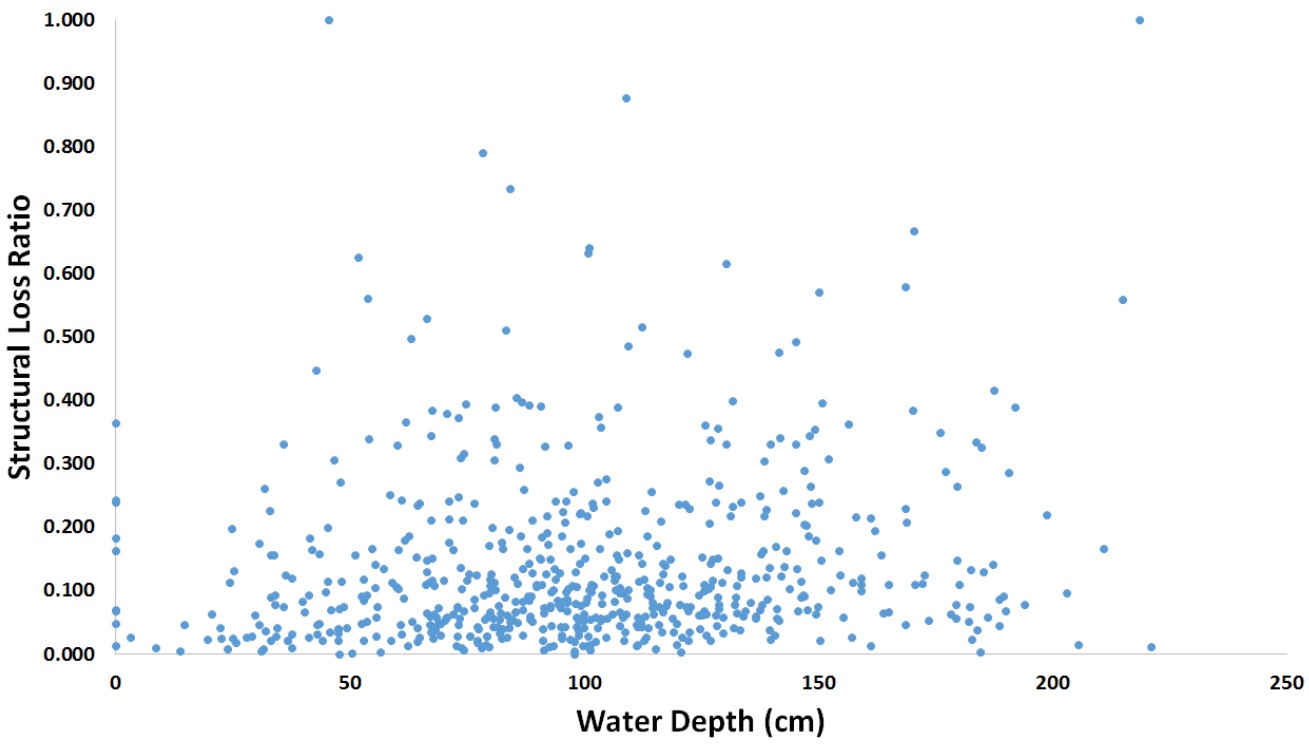

**Figure 3. Empirical data utilised for calibrating the FLF-IT model (613 relative damage records in the original dataset).**

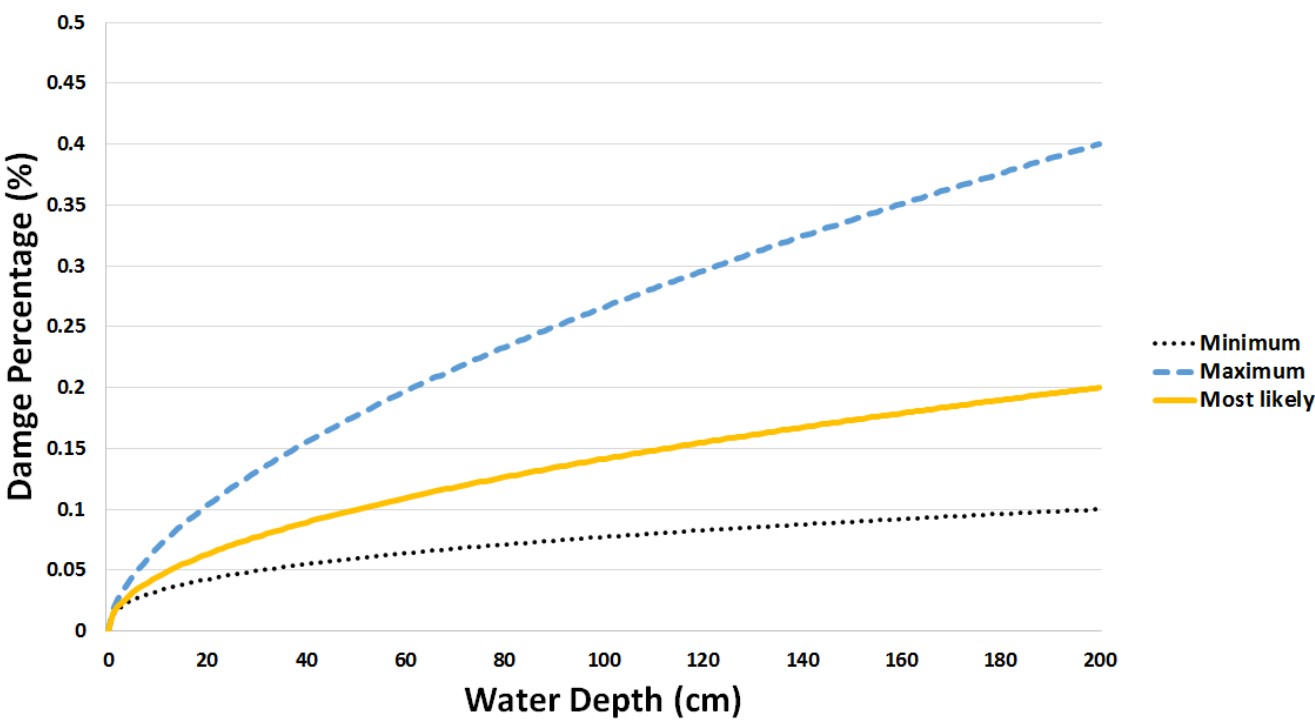

**Figure 4. Visualisation of minimum, most likely and maximum damage functions, calculated by bootstrap and chi-square test of goodness of fit.**

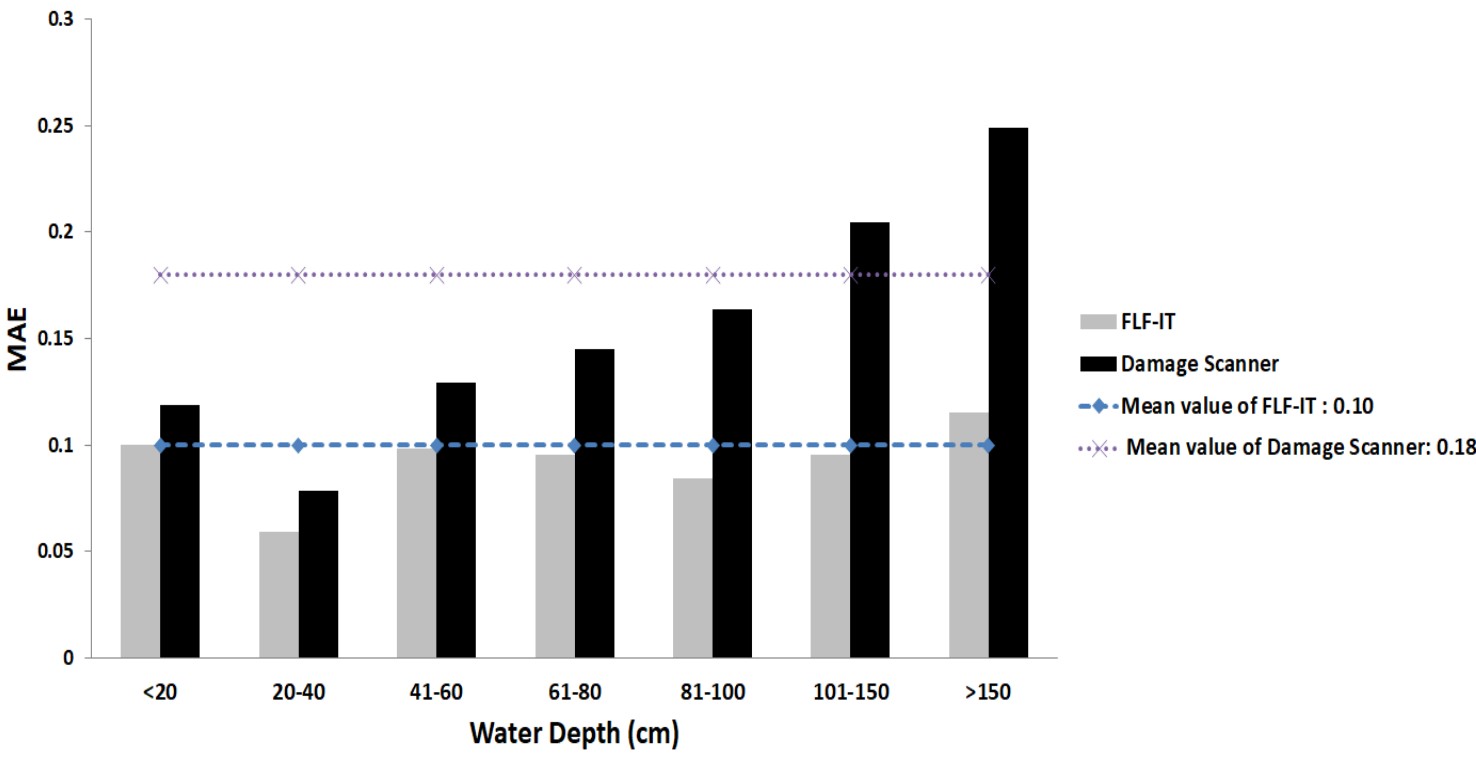

**Figure 5. Comparison of the flood damage estimation models' precision per water-depth class (MAE: mean absolute error; Number of damage records for each sub-class of water depth, respectively, are 14, 36, 52, 96, 125, 222, and 68)**

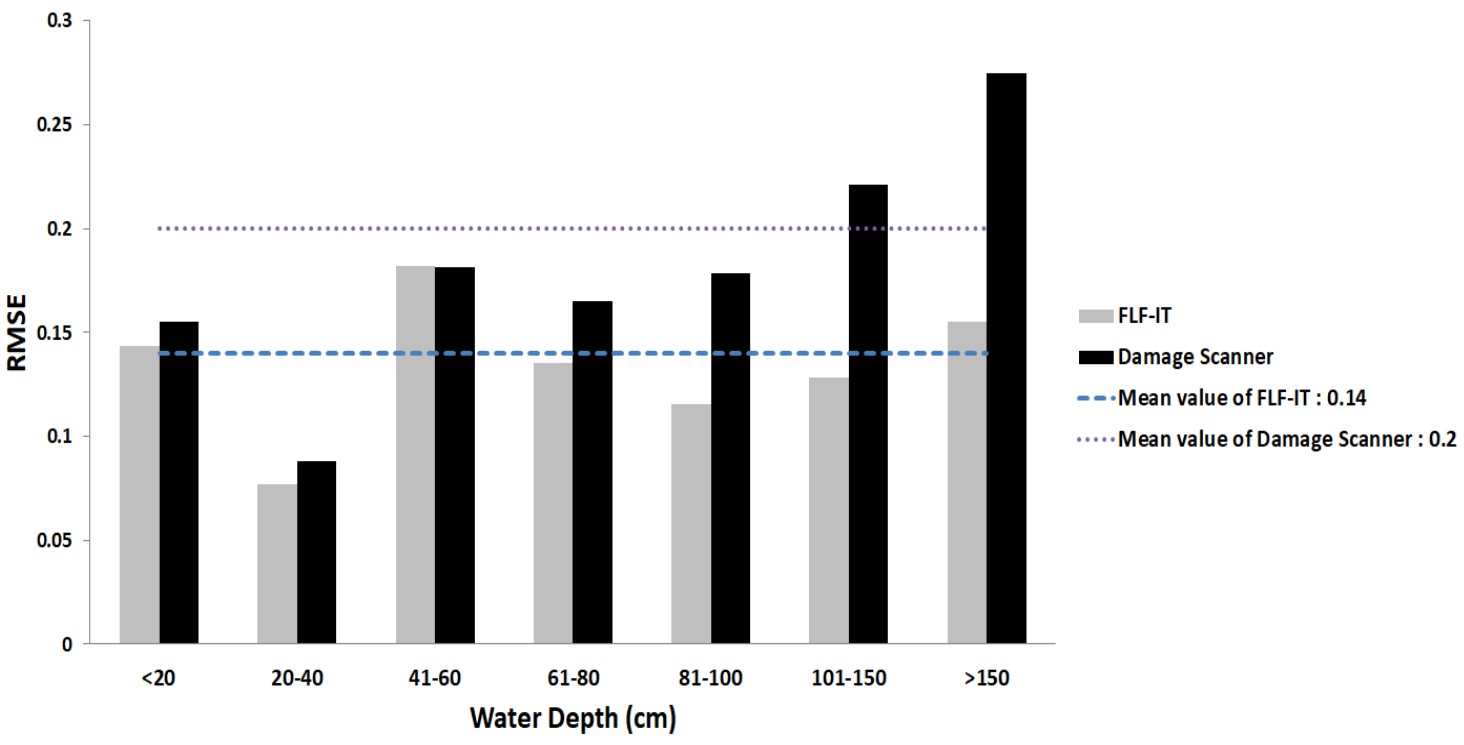

**Figure 6. Comparison of the flood damage estimation models' precision per water-depth class (RMSE: root mean square error; Number of samples for each sub-class of water depth, respectively, are 14, 36, 52, 96, 125, 222, and 68)**

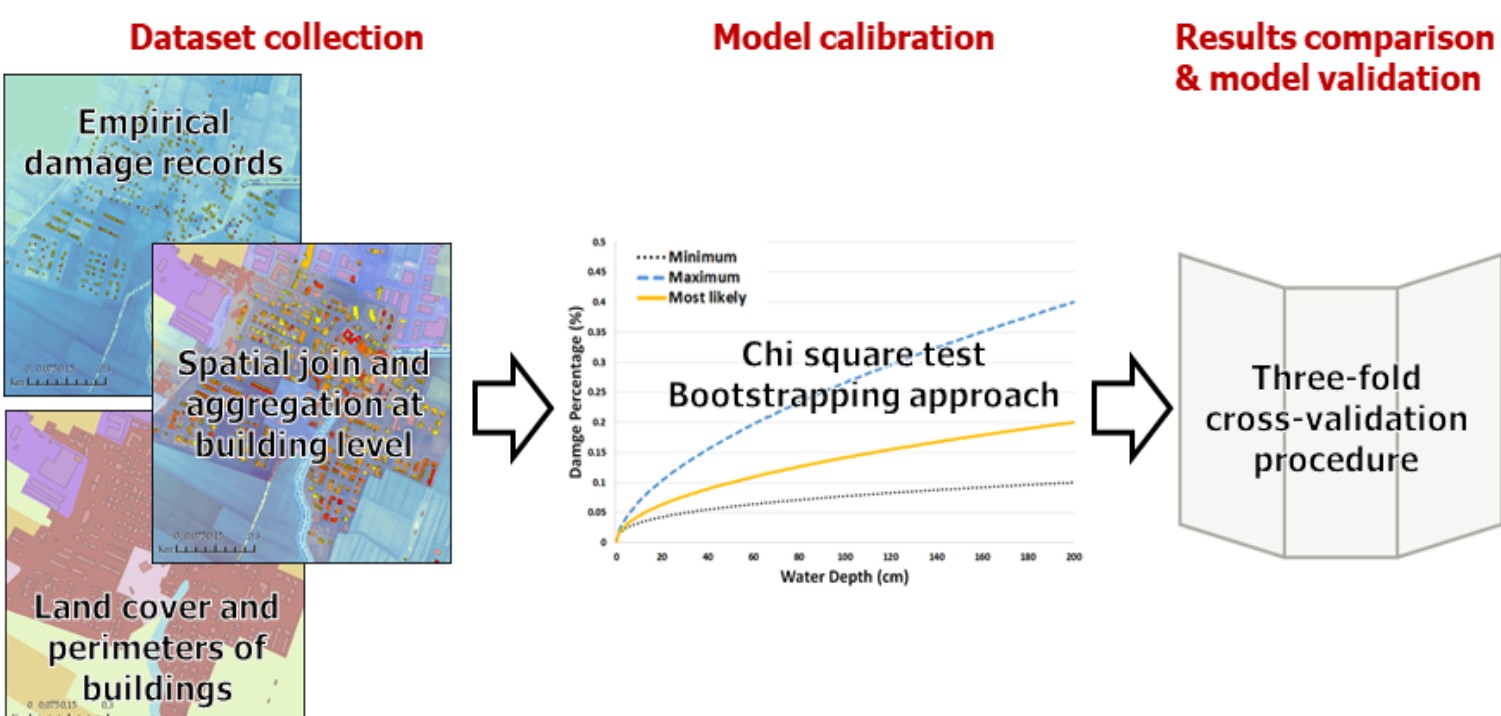

**Figure 7. Visualisation of the workflow and the methodological steps of the study**