# Peer review of "Flood loss modelling with FLF-IT: a new Flood Loss Function for Italian residential structures"

_Natural Hazards and Earth System Sciences, 2017_

## Referee Comment (RC1) · Anonymous Referee #1 · 4 Apr 2017

Summary

In this study a flood loss function was derived from a flood event in the Emilia-Romagna region in Northern Italy in January 2014. The flood loss function used in this study was developed by the first author of this manuscript using a case study from Australia and was calibrated to the case study area in Italy using empirical data. The data used for calibration comprises official damage records of three affected municipalities in Emilia-Romagna and the water depth that was modelled by a combination of 1-D and 2-D hydrological models. For the calibration of the function, bootstrap samples were used to find the best fitting values for two different parameters (root function and the maximum relative damage) based on a chi-square test of goodness of fit. The average of all

bootstrap rounds for the two parameters was used to calibrate the "most likely" function. In addition, the parameter sets that maximizes and minimizes the depth damage curve were used as minimum and maximum damage scenarios. A three-fold cross-validation was applied to validate the function with the same data set. For that, the model was calibrated in three iterations with leaving out a different third of the data for testing each time. The model performance for predicting the relative damage was evaluated using the MBE, MAE and RMSE. The MBE showed an overestimation for the first and an underestimation for the second and third iteration, leaving the average mean bias error at zero. The values for MAE and RMSE were ranging between 9 and 10% and 12 and 16% respectively. In a second step, the model was validated using absolute damage values. Therefore, the 95% confidence interval of the absolute damage was calculated by resampling the empiric damage values using bootstrapping. The performance of the loss function was accepted, when the predicted absolute damage was within the 95% confidence interval. This was the case for all three validation iterations as well as the sum of all iterations. The absolute damage was predicted by using the loss function to predict the relative damage and multiply it with the building value.

General comments

Although the application of depth-damage functions for economic flood loss estimation is quite frequently addressed in literature (see Penning-Rowsell et al. 2005 , Merz et al. 2010 and Hammond et al. 2015) the study at hand presents a new approach to calibrate a synthetic flood loss function with empiric damage data. The language of the manuscript is clear and understandable. However, major weaknesses in the documentation of the data as well as in the presentation of the calibration of the loss function, have a considerable effect on the replicability of the study. In addition, the presented results are not discussed or framed in the context of existing studies, which makes it difficult to see the advantage of the presented method in comparison to similar approaches. Therefore, I recommend accepting the manuscript only after major revisions.

Specific comments

Data description

In the documentation of the data used in the manuscript, several information are missing or not accurately described, which makes it difficult to fully understand each step of the analysis. An overview table of the empiric data used for the model calibration could help to get a better understanding of the data set in terms of distribution and sample size. It remains also unclear what building values were used to calculate the relative damage. In L8 on p.5 the author states to use "mean depreciated value" while in L13 p.5 it says "average market values". Values that represent the actual cost of the building based on material and labor can differ considerably from market values depending on the demand for housing in a certain area. In addition, the spatial matching of the damage values and building properties (L13-L17 on p. 5) should be outlined more clearly including Figure 2. This includes a description on how the damage records were aggregated on building level and which assumptions have been made in case damage records were not available for all units in a building. In Figure 2 the authors should explain what the points and building shapes mean and what we can learn from that.

Calibration and validation of FLF-IT

To avoid confusion, I would suggest moving the part that explains the cross-validation procedure (L12-14 on p.6) in front of the bootstrapping and calibration part (L24 on p. 5 to L6 on p.6) so it is in chronological order. It should also be stated how many samples were pulled out of the data set for each bootstrapping iteration. This is closely linked to the Data description section, where the overall size of the original dataset, the size of each subsample for cross-validation and the size of resampled dataset after bootstrapping should be stated. This can also help to explain the Number of samples in Table 1, which is unclear in the current version of the manuscript. Regarding the RMSE and MAE it should be stated if the percentage values are the original unit coming from the

relative damage or if the RMSE and MAE were normalized. In case the values were not normalized it is not possible to assess the predictive performance of the model without knowing the distribution of relative damage in the original dataset. Therefore, either the distribution of relative damage records in the original dataset should be provided or the RMSE and MAE should be normalized. In addition, I would recommend to slightly restructure Table 3 by showing the 95% confidence interval with the lower and upper boundaries in the second column instead of spreading it over column two and three.

Discussion

Given the fact that the application of depth-damage functions is a quite frequently addressed topic in flood research (see Merz et al. 2010 and Hammond et al. 2015), I would highly recommend to discuss the results of this manuscript in the framework of existing flood loss functions to highlight the unique and novel character of this study. This discussion should also include a critical evaluation of the study and the limitation of the study design. For example in L1 f. on p.8 the authors state that "Results of these validation tests illustrate the importance of model calibration, especially when the water depth is the only hydraulic parameter taken into account [...]." However, without the comparison with an uncalibrated function it is not possible to proof that predictions of calibrated loss functions are significantly better that uncalibrated ones. Since the loss function was calibrated on a single event in Italy using a single building type, the limitations in terms of a temporal and spatial transfer should be addressed as well.

Literature

P.2 L14: Jonkman (2007) provides a very detailed definition of (in)tangible and (in)direct flood damage and should be added here.

P.8 L4: Merz et al. (2013) and Schröter et al. (2014) showed that additional damage influencing factors considerably improve the damage predictions and therefore should be added here.

Technical corrections

P.1 L1: "Floods and storms": Damage caused by storms is actually not covered in this study. Therefore, I would recommend to include numbers for flood damage only.

P.2 L1 & P.3 L11f: "medium flood probability", "high flood probability". These are rather soft terms to describe flood probability. If available, I would recommend using numeric flood probabilities (e.g. "1% change to get flooded in any given year")

P.2 L17: "I-O models": write full name the first time a new term is mentioned

P.4 L10: "10 thousand": 10,000 or 10^4

P.4 L17: "125 mm of rain". Please provide timespan "e.g. 125 mm of rain in 48 hours"

P.4 L21 & L27f: "6.5 thousand hectares": convert into m^2 or km^2 to improve comparability with other values provided in this section.

P.4 L30: "bi-dimensional": 2-D

P.5 L1: "one-meter resolution": a one-meter resolution

Table 3: "(in EUR m)": Million? 10^6 EUR

P8. L21: "takes empirical data of damage and depth": According to the Data description section, the water depth was modelled and not empirically measured.

References

Hammond, M. J., Chen, A. S., Djordjević, S., Butler, D., & Mark, O. (2015). Urban flood impact assessment: A state-of-the-art review. Urban Water Journal, 12(1), 14-29.

Jonkman, S. N. (2007). Loss of life estimation in flood risk assessment; theory and applications. TU Delft, Delft University of Technology.

Merz, B., Kreibich, H., & Lall, U. (2013). Multi-variate flood damage assessment: a tree-based data-mining approach. Natural Hazards and Earth System Science, 13(1),

53-64.

Merz, B., Kreibich, H., Schwarze, R., & Thieken, A. (2010). Review article" Assessment of economic flood damage". Natural Hazards and Earth System Sciences, 10(8), 1697-1724.

Penning-Rowsell, E., Johnson, C., Tunstall, S., Tapsell, S., Morris, J., Chatterton, J., & Green, C. (2005). The benefits of flood and coastal risk management: a handbook of assessment techniques: Middlesex University Press.

Schröter, K., Kreibich, H., Vogel, K., Riggelsen, C., Scherbaum, F., & Merz, B. (2014). How useful are complex flood damage models? Water Resources Research, 50(4), 3378-3395.

---

## Referee Comment (RC2) · Anonymous Referee #2 · 10 Apr 2017

This paper aims to calibrate and validate a new relative flood loss function for Italian residential structures based on real damage data collected from a river flood event in the region of Emilia-Romagna at 2014. Authors are focusing on direct tangible damage, and the spatial scale is on the order of individual buildings. The function was developed based on an Australian approach (FLFA), which represents the confidence limits that exist around the parameterized functional depth-damage relationship. In the next step, the performance of the model was also validated for the prediction of loss ratios and absolute damage values. In this regard, a three-fold cross-validation procedure was carried out over the empirical sample to measure the range of uncertainty from the actual damage data. The validation procedure shows that the newly derived function

performs well and the results of these validation tests illustrate the importance of model calibration.

General comment

I have read the paper with great interest and the main objective addressed by the manuscript is framed appropriately to the scope of the journal. Overall, I think that the paper is well written, the results are nicely presented and the presented study could provide interesting empirical and quantitative insights. Nevertheless, some revisions are necessary to make a few points clearer and I recommend to accept it only after these revisions.

Specific comments

Materials and Methods part. In this part, I believe that the authors must use numbers rather than describing numbers with text (i.e. 10.000 kmˆ2 rather than 10 thousand kmˆ2). The methodology is well described and the method sounds scientifically correct but I believe and as it stated by another reviewer they should describe their methodological steps chronologically in order to avoid confusion. Additionally, I would suggest the authors to remove section 2 on the section describing their methodological steps in order to increase reader's friendliness. Moreover, I suggest the authors to give more information about the raw data used. As a reviewer without knowledge of the raw dataset, this is hard to assess. Please describe in more detail how total structure damage, average market value and mean water depth were calculated. On the data description part, change 'hydrological simulation' by 'hydraulic simulation' and 'bi-dimensional hydrological model' by '2D hydraulic model'.

Discussion. In general, the discussion part is missing apart a small discussion of their findings in section 4. I would suggest the authors to describe their results in more detail as well as with respect to findings from other case studies available in the literature. A more detailed comparison between the flood loss function for Italian residential structures presented in this study with other processes or other types of elements at

risk would be in my opinion an added value and would underline the importance of the specific one presented here.

Literature.

Karagiorgos K, Heiser M, Thaler T, Hübl J, Fuchs S. (2016) Micro-sized enterprises: vulnerability to flash floods. Nat Hazards 84: 1091-1107

Karagiorgos K, Thaler T, Heiser M, Hübl J, Fuchs S (2016) Integrated flash flood vulnerability assessment: insights from East Attica, Greece. J Hydrol. 541(A): 553-562

Fuchs S, Kuhlicke C, Meyer V (2011) Editorial for the special issue: vulnerability to natural hazards-the challenge of integration. Nat Hazards 58(2):609-619

Luino F, Cirio CG, Biddoccu M, Agangi A, Giulietto W, Godone F, Nigrelli G (2009) Application of a model to the evaluation of flood damage. Geoinformatica 13:339-353

Papathoma-Köhle M, Keiler M, Totschnig R, Glade T (2012) Improvement of vulnerability curves using data from extreme events: Debris flow event in South Tyrol. Nat Hazards 64(3):2083-2105

Totschnig R, Sedlacek W, Fuchs S (2011) A quantitative vulnerability function for fluvial sediment transport. Nat Hazards 58(2):681-703

---

## Author Comment (AC1) · 3 May 2017

**The authors wish to thank the editors and reviewers for their time and effort for reviewing our manuscript. We hope that the changes have improved the manuscript to a level that is suitable for publication, and we look forward to your response.**

**Reviewer 1**

**General comments**

Major weaknesses in the documentation of the data as well as in the presentation of the calibration of the loss function, have a considerable effect on the replicability of the study.
We appreciate your comment. In the new version, further explanations have been added. Please see the highlighted changes below.

In addition, the presented results are not discussed or framed in the context of existing studies, which makes it difficult to see the advantage of the presented method in comparison to similar approaches.
We are grateful for your suggestion. As mentioned below, a detailed comparison has been added to the new version. Please see section 5.

**Specific comments**

**Data description**

In the documentation of the data used in the manuscript, several information are missing or not accurately described, which makes it difficult to fully understand each step of the analysis.
Data description have been made more detailed and clear; suggested technical corrections have been implemented, and an explanation on how information is combined have been added, as explained in the next specific answers.

An overview table of the empiric data used for the model calibration could help to get a better understanding of the data set in terms of distribution and sample size.
Many thanks for your suggestion. In the new version, distribution and size of empirical data utilised for model calibration have been shown in *Fig. 3,* and it has been presented in the *caption of Figs. 5 and 6.*

It remains also unclear what building values were used to calculate the relative damage. In L8 on p.5 the author states to use "mean depreciated value" while in L13 p.5 it says "average market values". Values that represent the actual cost of the building based on material and labour can differ considerably from market values depending on the demand for housing in a certain area.
We are very grateful for your comment. The sentence has been amended.
*Please see L19 on p.4: "The recorded damage is compared to the average market values of the residential properties, as reported by the cadastral map for the semester preceding the flood event."*

In addition, the spatial matching of the damage values and building properties (L13-L17 on p. 5) should be outlined more clearly including Figure 2. This includes a description on how the damage records were aggregated on building level and which assumptions have been made in case damage records were not available for all units in a building. In Figure 2 the authors should explain what the points and building shapes mean and what we can learn from that.

The processing of raw data and the spatial aggregation process is now described in more detail. The Figure caption now explains in detail what the points and shapes are.
*Please see L24-29 on p4 and the caption of Figure 2.*

**Calibration and validation of FLF-IT**

To avoid confusion, I would suggest moving the part that explains the cross-validation procedure (L12 14 on p.6) in front of the bootstrapping and calibration part (L24 on p. 5 to L6 on p.6) so it is in chronological order.
We appreciate your suggestion. As a matter of fact, cross-validation procedure was related to the model validation which is one step after model calibration. In the new version, to avoid any confusion, calibration and validation parts are totally separated from each other.

It should also be stated how many samples were pulled out of the data set for each bootstrapping iteration. This is closely linked to the Data description section, where the overall size of the original dataset, the size of each subsample for cross-validation and the size of resampled dataset after bootstrapping should be stated. This can also help to explain the Number of samples in Table 1, which is unclear in the current version of the manuscript.
The overall size of the original dataset used for model calibration (613 samples) is presented in *L31 on p4, L3 on p6, Table 1, and the caption of Figure 3.*
Number of samples utilised in model validation are also added. *Please see L14-17 on p7*

Regarding the RMSE and MAE it should be stated if the percentage values are the original unit coming from the relative damage or if the RMSE and MAE were normalized. In case the values were not normalized it is not possible to assess the predictive performance of the model without knowing the distribution of relative damage in the original dataset. Therefore, either the distribution of relative damage records in the original dataset should be provided or the RMSE and MAE should be normalized.
We are very grateful for your comment. Distribution of the relative damage records is depicted in *Fig. 3*, and it is presented in the caption of *Figs. 5, and 6*.
It is also discussed in the highlighted parts of section 5.2

In addition, I would recommend to slightly restructure Table 3 by showing the 95% confidence interval with the lower and upper boundaries in the second column instead of spreading it over column two and three.
Corrected. *Please see Table 3.*

**Discussion**

Given the fact that the application of depth-damage functions is a quite frequently addressed topic in flood research (see Merz et al. 2010 and Hammond et al. 2015), I would highly recommend to discuss the results of this manuscript in the framework of existing flood loss functions to highlight the unique and novel character of this study. This discussion should also include a critical evaluation of the study and the limitation of the study design. For example in L1 f. on p.8 the authors state that "Results of these validation tests illustrate the importance of model calibration, especially when the water depth is the only hydraulic parameter taken into account [: : :]." However, without the comparison with an uncalibrated function it is not possible to proof that predictions of calibrated loss functions are significantly better that uncalibrated ones. Since the loss function was calibrated on a single event in Italy using a single building type, the limitations in terms of a temporal and spatial transfer should be addressed as well.
We appreciate your suggestion. In the new version, a detailed comparison has been added, and the limitations have been discussed. In this version, section 5 which is related to results comparison and model validation has been changed substantially.

*Please see the highlighted parts of section 5.*

**Literature**

P.2 L14: Jonkman (2007) provides a very detailed definition of (in)tangible and (in)direct flood damage and should be added here.
Added.
*Please see L16 on p.2*

P.8 L4: Merz et al. (2013) and Schröter et al. (2014) showed that additional damage influencing factors considerably improve the damage predictions and therefore should be added here.
Added.
*Please see L25 on p.8*

**Technical corrections**

P.1 L1: "Floods and storms": Damage caused by storms is actually not covered in this study. Therefore, I would recommend to include numbers for flood damage only.
Corrected. Now it refers to floods only. The following numbers were already related to flood inundation.
*Please see L29 on p.1*

P.2 L1 & P.3 L11f: "medium flood probability", "high flood probability". These are rather soft terms to describe flood probability. If available, I would recommend using numeric flood probabilities (e.g. "1% change to get flooded in any given year")
Corrected. They have been changed to probability in terms of return period.
*Please see L2 on p.2; & L22 and L23 on p.3.: "exposed to a flood probability of once every 100 to 200 years" and "return period between xxx and xxx years".*

P.2 L17: "I-O models": write full name the first time a new term is mentioned
Corrected as "Input-Output models".
*Please see L20 on p.2*

P.4 L10: "10 thousand": 10,000 or 10^4
Changed to "10,000".
*Please see L21 on p.3*

P.4 L17: "125 mm of rain". Please provide timespan "e.g. 125 mm of rain in 48 hours"
Corrected.
*Please see L29 on p.3: "with an areal mean of 125 mm of cumulated rain over 72 hours flowing in the Secchia catchment."*

P.4 L21 & L27f: "6.5 thousand hectares": convert into m^2 or km^2 to improve comparability with other values provided in this section.
Done.
*Please see L2 & L9 on p.4*

P.4 L30: "bi-dimensional": 2-D
Done.
*Please see L10 on p.4*

P.5 L1: "one-meter resolution": a one-meter resolution
Done.
*Please see L11 on p.4*

Table 3: "(in EUR m)": Million? 10ˆ6 EUR
Corrected.
*Please see Table 3.*

P8. L21: "takes empirical data of damage and depth": According to the Data description section, the water depth was modelled and not empirically measured.
This sentence was related to FLFA and not FLF-IT. However, in order to avoid any confusion, the sentence was amended.
*Please see L3 on p.10:  "The FLFA approach takes data of damage and depth."*

**References**

Hammond, M. J., Chen, A. S., Djordjevi´c, S., Butler, D., & Mark, O. (2015). Urban flood impact assessment: A state-of-the-art review. Urban Water Journal, 12(1), 14-29.
Added.
*Please see L10 on p.2*

Jonkman, S. N. (2007). Loss of life estimation in flood risk assessment; theory and applications. TU Delft, Delft University of Technology.
Added.
*Please see L16 on p.2*

Merz, B., Kreibich, H., & Lall, U. (2013). Multi-variate flood damage assessment: a tree-based data-mining approach. Natural Hazards and Earth System Science, 13(1), 53-64.
Added.
*Please see L25 on p.8*

Merz, B., Kreibich, H., Schwarze, R., & Thieken, A. (2010). Review article" Assessment of economic flood damage". Natural Hazards and Earth System Sciences, 10(8), 1697-
1724.
It is already there.
*Please see L13, 24, and 31 on p.2*

Penning-Rowsell, E., Johnson, C., Tunstall, S., Tapsell, S., Morris, J., Chatterton, J., & Green, C. (2005). The benefits of flood and coastal risk management: a handbook of assessment techniques: Middlesex University Press.
Added.
*Please see L13 on p.2*

Schröter, K., Kreibich, H., Vogel, K., Riggelsen, C., Scherbaum, F., & Merz, B. (2014). How useful are complex flood damage models? Water Resources Research, 50(4), 3378-3395.
Added.
*Please see L25 on p.8*

---

## Author Comment (AC2) · 3 May 2017

**Reviewer 2**

**The authors wish to thank the editors and reviewers for their time and effort for reviewing our manuscript. We hope that the changes have improved the manuscript to a level that is suitable for publication, and we look forward to your response.**

**Specific comments**

**Materials and Methods**

In this part, I believe that the authors must use numbers rather than describing numbers with text (i.e. 10.000 kmˆ2 rather than 10 thousand kmˆ2).
Corrected.
*Please see L21 on p.3 and L2 & L9 on p.4*

The methodology is well described and the method sounds scientifically correct but I believe and as it stated by another reviewer they should describe their methodological steps chronologically in order to avoid confusion. Additionally, I would suggest the authors to remove section 2 on the section describing their methodological steps in order to increase reader's friendliness.
Many thanks for the comment. To avoid any confusion, section 2 (explanation about the FLFA method) has been moved to before the model calibration part. Also, the "Model Calibration" and the "Model Validation" parts are totally separated from each other.

Moreover, I suggest the authors to give more information about the raw data used. As a reviewer without knowledge of the raw dataset, this is hard to assess. Please describe in more detail how total structure damage, average market value and mean water depth were calculated.
We are grateful for your suggestion. The processing of raw data and the spatial aggregation process is now described in more detail.
*Please see L24-29 on p4 and the caption of figure 2.*

On the data description part, change 'hydrological simulation' by 'hydraulic simulation' and 'bi-dimensional hydrological model' by '2D hydraulic model'.
Corrected.
*Please see L7&10 on p.4.*

**Discussion**

In general, the discussion part is missing apart a small discussion of their findings in section 4. I would suggest the authors to describe their results in more detail as well as with respect to findings from other case studies available in the literature. A more detailed comparison between the flood loss function for Italian residential structures presented in this study with other processes or other types of elements at risk would be in my opinion an added value and would underline the importance of the specific one presented here.
We appreciate your suggestion. In the new version, a detailed comparison has been added, and the results are discussed in more details. In this version, *section 5* which is related to results comparison and model validation has been changed substantially.
*Please see the highlighted parts of section 5.*

**Literature**.

Karagiorgos K, Heiser M, Thaler T, Hübl J, Fuchs S. (2016) Micro-sized enterprises: vulnerability to flash floods. Nat Hazards 84: 1091-1107
Added.
*Please see L17 on p.7*

Karagiorgos K, Thaler T, Heiser M, Hübl J, Fuchs S (2016) Integrated flash flood vulnerability assessment: insights from East Attica, Greece. J Hydrol. 541(A): 553-562
Added.
*Please see L24 on p.7*

Fuchs S, Kuhlicke C, Meyer V (2011) Editorial for the special issue: vulnerability to natural hazards-the challenge of integration. Nat Hazards 58(2):609-619
Added.
*Please see L24 on p.8*

Luino F, Cirio CG, Biddoccu M, Agangi A, Giulietto W, Godone F, Nigrelli G (2009) Application of a model to the evaluation of flood damage. Geoinformatica 13:339-353
Added.
*Please see L11 on p.3*

Papathoma-Köhle M, Keiler M, Totschnig R, Glade T (2012) Improvement of vulnerability curves using data from extreme events: Debris flow event in South Tyrol. Nat Hazards 64(3):2083-2105
Added.
*Please see L12 on p.3*

Totschnig R, Sedlacek W, Fuchs S (2011) A quantitative vulnerability function for fluvial sediment transport. Nat Hazards 58(2):681-703
Added.
*Please see L24 on p.2*

---

## Author Response (AR1)

**The authors wish to thank the editors and reviewers for their time and effort for reviewing our manuscript. We hope that the changes have improved the manuscript to a level that is suitable for publication, and we look forward to your response.**

**Reviewer 1**

**General comments**

Major weaknesses in the documentation of the data as well as in the presentation of the calibration of the loss function, have a considerable effect on the replicability of the study.
We appreciate your comment. In the new version, further explanations have been added. Please see the highlighted changes below.

In addition, the presented results are not discussed or framed in the context of existing studies, which makes it difficult to see the advantage of the presented method in comparison to similar approaches.
We are grateful for your suggestion. As mentioned below, a detailed comparison has been added to the new version.
*Please see the changes in section 5.*

**Specific comments**

**Data description**

In the documentation of the data used in the manuscript, several information are missing or not accurately described, which makes it difficult to fully understand each step of the analysis.
As explained in the next specific answers, data description has been made more detailed and clear; suggested technical corrections have been implemented, and an explanation on how information is combined has been added.

An overview table of the empiric data used for the model calibration could help to get a better understanding of the data set in terms of distribution and sample size.
Many thanks for your suggestion. In the new version, distribution and size of the empirical data utilised for model calibration have been shown in *Fig. 3*. It has also been presented in the *caption of Figs. 5 and 6.*

It remains also unclear what building values were used to calculate the relative damage. In L8 on p.5 the author states to use "mean depreciated value" while in L13 p.5 it says "average market values". Values that represent the actual cost of the building based on material and labour can differ considerably from market values depending on the demand for housing in a certain area.
We are very grateful for your comment. The sentence has been amended.
*Please see L25 on p.4: "The recorded damage is compared to the average market values of the residential properties, as reported by the cadastral map for the semester preceding the flood event."*

In addition, the spatial matching of the damage values and building properties (L13-L17 on p. 5) should be outlined more clearly including Figure 2. This includes a description on how the damage records were aggregated on building level and which assumptions have been made in case damage records were not available for all units in a building. In Figure 2 the authors should explain what the points and building shapes mean and what we can learn from that.

*The processing of raw data and the spatial aggregation process is now described in more detail. The Figure caption now explains in detail what the points and shapes are.*
*Please see L30 on p4- L3 on p5 and the caption of Figure 2.*

**Calibration and validation of FLF-IT**

To avoid confusion, I would suggest moving the part that explains the cross-validation procedure (L12 14 on p.6) in front of the bootstrapping and calibration part (L24 on p. 5 to L6 on p.6) so it is in chronological order.

*We appreciate your suggestion. As a matter of fact, cross-validation procedure was related to the model validation which is one step after model calibration. In the new version, to avoid any confusion, the model calibration and the model validation parts are totally separated from each other.*

It should also be stated how many samples were pulled out of the data set for each bootstrapping iteration. This is closely linked to the Data description section, where the overall size of the original dataset, the size of each subsample for cross-validation and the size of resampled dataset after bootstrapping should be stated. This can also help to explain the Number of samples in Table 1, which is unclear in the current version of the manuscript.

*The overall size of the original dataset used for model calibration (613 samples) is presented in L5 on p5, L10 on p6, Table 1, and the caption of Figure 3.*
*Number of samples utilised in model validation are also added. Please see L22-25 on p7*

Regarding the RMSE and MAE it should be stated if the percentage values are the original unit coming from the relative damage or if the RMSE and MAE were normalized. In case the values were not normalized it is not possible to assess the predictive performance of the model without knowing the distribution of relative damage in the original dataset. Therefore, either the distribution of relative damage records in the original dataset should be provided or the RMSE and MAE should be normalized.

*We are very grateful for your comment. Distribution of the relative damage records is depicted in Fig. 3, and it is presented in the caption of Figs. 5, and 6.*
*The distribution of damage ratios and the magnitude of errors according to some sub-classes of water depth is also discussed in section 5.2*

In addition, I would recommend to slightly restructure Table 3 by showing the 95% confidence interval with the lower and upper boundaries in the second column instead of spreading it over column two and three.

*Corrected. Please see Table 3.*

**Discussion**

Given the fact that the application of depth-damage functions is a quite frequently addressed topic in flood research (see Merz et al. 2010 and Hammond et al. 2015), I would highly recommend to discuss the results of this manuscript in the framework of existing flood loss functions to highlight the unique and novel character of this study. This discussion should also include a critical evaluation of the study and the limitation of the study design. For example in L1 f. on p.8 the authors state that "Results of these validation tests illustrate the importance of model calibration, especially when the water depth is the only hydraulic parameter taken into account [: : :]." However, without the comparison with an uncalibrated function it is not possible to proof that predictions of calibrated loss functions are significantly better that uncalibrated ones. Since the loss function was calibrated on a single event in Italy using a single building type, the limitations in terms of a temporal and spatial transfer should be addressed as well.

We appreciate your suggestion. In the new version, a detailed comparison has been added, the unique and novel characters of this model have been discussed, and the limitations of this study have been mentioned. In this version, section 5 which is related to results comparison and model validation has been changed substantially.
*Please see the highlighted parts of section 5.*

Furthermore, the novel characters of this model were mentioned before in *L27 on p1 & L17 on p10*. The limitations were also mentioned before in *L32 on p10*

**Literature**

P.2 L14: Jonkman (2007) provides a very detailed definition of (in)tangible and (in)direct flood damage and should be added here.
Added.
*Please see L19 on p.2*

P.8 L4: Merz et al. (2013) and Schröter et al. (2014) showed that additional damage influencing factors considerably improve the damage predictions and therefore should be added here.
Added.
*Please see L9 on p.9*

**Technical corrections**

P.1 L1: "Floods and storms": Damage caused by storms is actually not covered in this study. Therefore, I would recommend to include numbers for flood damage only.
Corrected. Now it refers to floods only. The following numbers were already related to flood inundation.
*Please see L2 on p.2*

P.2 L1 & P.3 L11f: "medium flood probability", "high flood probability". These are rather soft terms to describe flood probability. If available, I would recommend using numeric flood probabilities (e.g. "1% change to get flooded in any given year")
Corrected. They have been changed to probability in terms of return period.
*Please see L5 on p.2; & L26 and L27 on p.3.: "exposed to a flood probability of once every 100 to 200 years" and "return period between xxx and xxx years".*

P.2 L17: "I-O models": write full name the first time a new term is mentioned
Corrected as "Input-Output models".
*Please see L23 on p.2*

P.4 L10: "10 thousand": 10,000 or 10^4
Changed to "10,000".
*Please see L25 on p.3*

P.4 L17: "125 mm of rain". Please provide timespan "e.g. 125 mm of rain in 48 hours"
Corrected.
*Please see L4 on p.4: "with an areal mean of 125 mm of cumulated rain over 72 hours flowing in the Secchia catchment."*

P.4 L21 & L27f: "6.5 thousand hectares": convert into m^2 or km^2 to improve comparability with other values provided in this section.
Done.

*Please see L8 & L15 on p.4*

P.4 L30: "bi-dimensional": 2-D
Done.
*Please see L16 on p.4*

P.5 L1: "one-meter resolution": a one-meter resolution
Done.
*Please see L17 on p.4*

Table 3: "(in EUR m)": Million? 10ˆ6 EUR
Corrected.
*Please see Table 3.*

P8. L21: "takes empirical data of damage and depth": According to the Data description section, the water depth was modelled and not empirically measured.
This sentence was related to FLFA and not FLF-IT. However, in order to avoid any confusion, the sentence was amended.
*Please see L13 on p.10: "The FLFA approach takes data of damage and depth."*

We very much appreciate your suggestions. Additional references were provided in the revised manuscript. Please see the highlighted references.

**Reviewer 2**

**The authors wish to thank the editors and reviewers for their time and effort for reviewing our manuscript. We hope that the changes have improved the manuscript to a level that is suitable for publication, and we look forward to your response.**

**Specific comments**

**Materials and Methods**

In this part, I believe that the authors must use numbers rather than describing numbers with text (i.e. 10.000 kmˆ2 rather than 10 thousand kmˆ2).
Corrected.
*Please see L25 on p.3 and L8 & L15 on p.4*

The methodology is well described and the method sounds scientifically correct but I believe and as it stated by another reviewer they should describe their methodological steps chronologically in order to avoid confusion. Additionally, I would suggest the authors to remove section 2 on the section describing their methodological steps in order to increase reader's friendliness.
Many thanks for the comment. To avoid any confusion, section 2 (explanation about the FLFA method) has been moved to before the model calibration part. Also, the "Model Calibration" and the "Model Validation" parts are totally separated from each other.

Moreover, I suggest the authors to give more information about the raw data used. As a reviewer without knowledge of the raw dataset, this is hard to assess. Please describe in more detail how total structure damage, average market value and mean water depth were calculated.
We are grateful for your suggestion. The processing of raw data and the spatial aggregation process is now described in more detail.
*Please see L30 on p.4 - L3 on p.5 and the caption of figure 2.*

On the data description part, change 'hydrological simulation' by 'hydraulic simulation' and 'bi-dimensional hydrological model' by '2D hydraulic model'.
Corrected.
*Please see L13 on p.4 & L16 on p.4*

**Discussion**

In general, the discussion part is missing apart a small discussion of their findings in section 4. I would suggest the authors to describe their results in more detail as well as with respect to findings from other case studies available in the literature. A more detailed comparison between the flood loss function for Italian residential structures presented in this study with other processes or other types of elements at risk would be in my opinion an added value and would underline the importance of the specific one presented here.
We appreciate your suggestion. In the new version, a detailed comparison has been added, and the results are discussed in more details. In this version, *section 5* which is related to results comparison and model validation has been changed substantially.
*Please see the highlighted parts in section 5.*

**Literature**.

Karagiorgos K, Heiser M, Thaler T, Hübl J, Fuchs S. (2016) Micro-sized enterprises: vulnerability to flash floods. Nat Hazards 84: 1091-1107

Karagiorgos K, Thaler T, Heiser M, Hübl J, Fuchs S (2016) Integrated flash flood vulnerability assessment: insights from East Attica, Greece. J Hydrol. 541(A): 553-562

Fuchs S, Kuhlicke C, Meyer V (2011) Editorial for the special issue: vulnerability to natural hazards-the challenge of integration. Nat Hazards 58(2):609-619

Luino F, Cirio CG, Biddoccu M, Agangi A, Giulietto W, Godone F, Nigrelli G (2009) Application of a model to the evaluation of flood damage. Geoinformatica 13:339-353

Papathoma-Köhle M, Keiler M, Totschnig R, Glade T (2012) Improvement of vulnerability curves using data from extreme events: Debris flow event in South Tyrol. Nat Hazards 64(3):2083-2105

Totschnig R, Sedlacek W, Fuchs S (2011) A quantitative vulnerability function for fluvial sediment transport. Nat Hazards 58(2):681-703

We very much appreciate your suggestions. Additional references were provided in the revised manuscript. Please see the highlighted references.

[revised manuscript text omitted]

---

## Author Response (AR2)

**The authors wish to thank the editor and reviewers for their time and effort for reviewing our manuscript. We hope that the changes have improved the manuscript to a level that is suitable for publication, and we look forward to your response.**

**Reviewer 1**

**General comments**

The revised version of the manuscript significantly improved compared to the previous version. The methodology as well as the data set is now explicitly described and splitting the model calibration in validation in two chapters makes it easier for the reader to follow the study design.

Thank you for your comments. The minor corrections have been implemented as suggested.

P. 5 L1-3: It is unclear why the authors in a first step use geo-referencing to transfer the address into a point coordinate to visually inspect if the address is within the building footprint. As a person not familiar with the Italian house number system, I would assume that every damage entry for an address (incl. house number) also belongs to a particular building. Please provide reasons for your approach.

The motivation for this procedure is now stated more clearly in the related paragraph (P4 L31 – P5 L5). Also, please see P5 L3: "This spatial join is necessary since building perimeters do not include any information about addresses."

P6 L10-20, P.7 L21-26: According to P.7 L22-25 the authors are using a 3-fold cross validation, where for each of the three iterations two thirds of the data is used for calibration and one third is used for validation. However, reading the previous chapter P.6 L10-20, one could think that the full 613 observations are used for calibration. I would recommend mentioning the data splitting procedure in P.6 and that in fact three models are calibrated with the three different data sets. It is also not 100% clear to me, if the bootstrap approach described on P. 6, is performed before or after splitting the dataset for cross validation.

Model calibration (P.6) and model validation (P. 7-9) procedures are totally separated from each other.

As we have mentioned in "*P6 L10; Table 1; and Figure 3*", the entire original dataset including 613 relative damage records has been used for the model calibration and the related bootstrapping approach presented in Section 4 (P.6 L10-20).

After model calibration (as we have separated the calibration and the validation parts totally), we need to test the performance of the method and fulfil a model validation. For model validation using a 3-fold cross validation technique, we have to partition the original dataset into three equally sized folders and calibrate and test the model for three times (here, for each iteration, two-thirds of the data are used for calibration, and one-third is used for validation). Eventually, results would be averaged over all three iterations (as mentioned in P7 L22-29; Table 2; and Table 3).

**Technical Corrections**

Abstract L1: amount in million Euros instead of dollars

Corrected to Euros. Please see P.1 L13.

P. 3 L26: 2.5 thousand to 25,000

Corrected (2,500). Please see P. 3 L26

P. 9 L17 -22: can be "made" instead of "found". Shorter sentences

Corrected as suggested. Please see P.9 L22-23.

P. 10 L15: "maximum damage as a percentage and the starting elevation for damage": please specify percentage of …; please clarify "starting elevation damage"

Corrected. Please see P.10 L15: "maximum damage as a percentage of the total building value, and the elevation of water which building start damaging."

Fig 5 and 6: Class limits of third class are probably wrong 14-60 should probably 41-60

Corrected. Please see Figs. 5 and 6.

**Reviewer 2**

Most of my concerns about the paper were addressed and the revised paper is appropriate to NHESS journal. In this line, it can be now accepted for publication with some minor revisions. In these revisions, the authors should re-check the paper for some typos (e.g. page 4/ line 4 "125 mm of cumulated") as well as to use symbols for the units (e.g. page 4/ line 7 "200 cubic meters per second"). Additionally, I suggest them - if it is possible - to create a workflow with their methodological steps to increase reader's friendliness.

We appreciate your comments and suggestions.

Units and typos have been fixed. Also, a figure which illustrates the workflow of the analysis has been added (Please see P.9 L5 & Fig. 7).

[revised manuscript text omitted]